**RESEARCH**

# Transcriptome- and proteome-wide association studies nominate determinants of kidney function and damage

Pascal Schlosser[1*] , Jingning Zhang[2], Hongbo Liu[3], Aditya L. Surapaneni[4,5], Eugene P. Rhee[6], Dan E. Arking[7], Bing Yu[8], Eric Boerwinkle[8,9], Paul A. Welling[10,11], Nilanjan Chatterjee[2], Katalin Susztak[3], Josef Coresh[1] and Morgan E. Grams[1,5]

*Correspondence:
pschlos3@jhu.edu

[1] Department of Epidemiology, Johns Hopkins Bloomberg School of Public Health, Baltimore, MD, USA
Full list of author information is available at the end of the article

## Abstract

**Background:** The pathophysiological causes of kidney disease are not fully understood. Here we show that the integration of genome-wide genetic, transcriptomic, and proteomic association studies can nominate causal determinants of kidney function and damage.

**Results:** Through transcriptome-wide association studies (TWAS) in kidney cortex, kidney tubule, liver, and whole blood and proteome-wide association studies (PWAS) in plasma, we assess for effects of 12,893 genes and 1342 proteins on kidney filtration (glomerular filtration rate (GFR) estimated by creatinine; GFR estimated by cystatin C; and blood urea nitrogen) and kidney damage (albuminuria). We find 1561 associations distributed among 260 genomic regions that are supported as putatively causal. We then prioritize 153 of these genomic regions using additional colocalization analyses. Our genome-wide findings are supported by existing knowledge (animal models for *MANBA, DACH1, SH3YL1, INHBB*), exceed the underlying GWAS signals (28 region-trait combinations without significant GWAS hit), identify independent gene/protein-trait associations within the same genomic region (*INHBC, SPRYD4*), nominate tissues underlying the associations (tubule expression of *NRBP1*), and distinguish markers of kidney filtration from those with a role in creatinine and cystatin C metabolism. Furthermore, we follow up on members of the TGF-beta superfamily of proteins and find a prognostic value of INHBC for kidney disease progression even after adjustment for measured glomerular filtration rate (GFR).

**Conclusion:** In summary, this study combines multimodal, genome-wide association studies to generate a catalog of putatively causal target genes and proteins relevant to kidney function and damage which can guide follow-up studies in physiology, basic science, and clinical medicine.

**Keywords:** Genetics, Nephrology, Chronic kidney disease, Transcriptomics, Proteomics, GWAS, TWAS, PWAS, eGFR, BUN, ACR, End-stage kidney disease

## Background

Chronic kidney disease (CKD), defined as a persistent decrement in glomerular filtration rate (GFR) or the presence of kidney damage, affects more than 10% of the adult population worldwide [1, 2]. GFR is typically estimated with serum creatinine (eGFRcr), serum cystatin C (eGFRcys), or blood urea nitrogen (BUN), and damage is quantified by the urinary albumin-to-creatinine ratio (ACR) [3]. Treatment for CKD is limited, in part, because the pathophysiological mechanisms contributing to its origin and progression are not fully understood. Gaining insight into disease pathogenesis can help identify new targets for pharmaceutical interventions.

Technical advances in transcriptomic and proteomic profiling provide unprecedented access to tissue-specific and circulating gene products which are relevant in disease and health. However, observational studies relating gene transcript or protein abundance directly to phenotypes can be confounded or represent reverse causation [4]. The integration of genetic data with gene transcripts, proteins, and phenotypes can help nominate specific biomarkers as potential causal determinants of disease [5]. For example, Gusev et al. [6] used single-nucleotide polymorphisms (SNPs) in *cis* to genetically impute transcripts and relate them to GWAS summary statistics of a trait of interest in order to provide evidence of causality (transcriptome-wide association study, or TWAS) [7–10]. We recently adapted this approach for proteome-wide association studies (PWAS) based on genetically imputed models of the plasma proteome (SomaScan V4 platform) [11]. In these implementations, TWAS and PWAS can be viewed as instrumental variable (IV) analyses akin to two-sample Mendelian randomization (MR) [12]. The genetic models used as instrumental variables are restricted to the *cis*-region of the gene transcript or protein, i.e., genetic variants within or close to the encoding gene, which reduces the risk of confounding by horizontal pleiotropy (independent of the protein). Combined with colocalization analyses accounting for multiple causal variants with the same region, these methods reduce the risk of genetic confounding due to linkage disequilibrium [13–15].

In order to implicate new roles of gene transcripts and proteins in the development of CKD, we apply the TWAS, PWAS, and colocalization methods to recent GWAS of kidney filtration and damage [16–18] (12,893 genes and 1342 proteins tested; GWAS N up to 1 million; Fig. 1). Essential tissues related to kidney function and damage are the *kidney cortex*, the part of the kidney where glomeruli interact with *(whole) blood*; the *kidney tubule* that participates in the exchange of substances from the tubular fluid (pre-urine) and blood; and the *liver* which has been shown to be co-regulated with kidney tissue [16], is linked to genetic determinants of eGFRcr [19], and which together with kidney are the key organs underlying the regulation of urinary metabolite concentrations [20]; hence, we focus the TWAS on these tissues. The PWAS focused on whole blood plasma, which represents the most targetable biospecimen. To increase confidence that a marker represents CKD processes rather than a metabolic byproduct, we integrate across different markers of filtration (eGFRcr, eGFRcys, and BUN) and kidney damage (ACR). To better isolate the location of putative causal genetic variants and the tissues in which they influence disease, we conduct conditional analyses to prioritize omics layers. Finally, to assess their feasibility as treatment targets, we pharmacologically annotate the markers of interest with approved drugs and drugs under development.

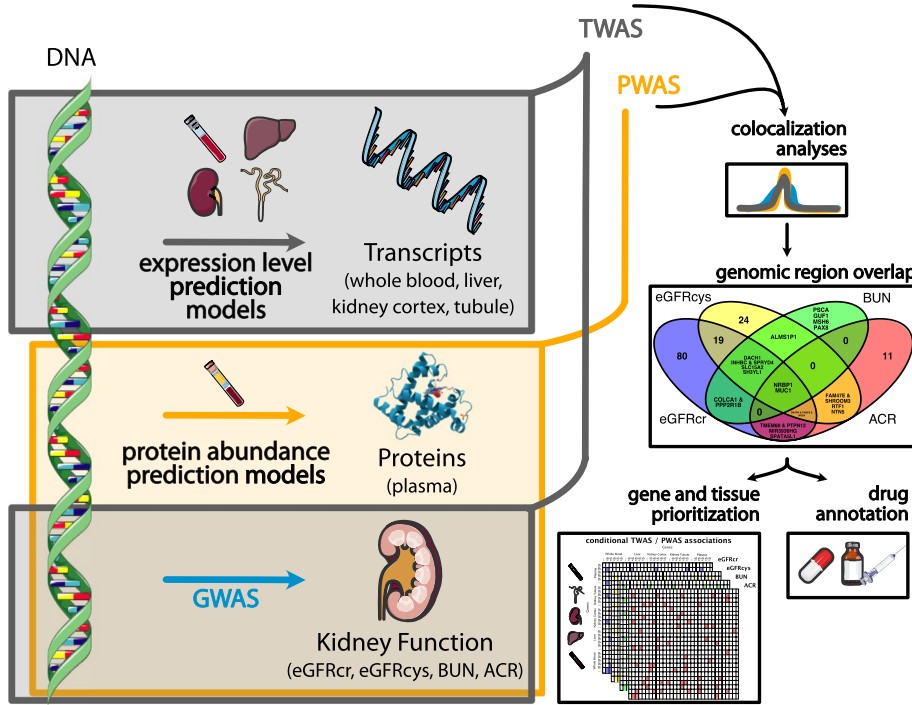

**Fig. 1** Workflow of integrated transcriptome-wide and proteome-wide association studies of kidney function and damage. We performed TWAS (gray boxes) and PWAS (orange box) using genetic instruments to model life-long differences in transcript expression and protein abundance and their effect on kidney function and damage (eGFRcr, eGFRcys, BUN, and ACR). Significant TWAS / PWAS associations which additionally displayed statistical colocalization of the kidney function / damage GWAS and the transcript / protein GWAS (eQTLs, pQTLs) were moved forward and compared across genomic regions and kidney function traits. Conditional analyses were used to prioritize genes and tissues of origin per genomic region. Putative treatment targets were pharmacologically annotated. Icon credit: Servier Medical Art by Servier (licensed under a Creative Commons Attribution 3.0 Unported License)

## Results

### Tissue-specific transcriptome-wide association studies (TWAS) of kidney function and damage

We conducted tissue-specific TWAS for four kidney function-related tissues, using models for kidney cortex (models built based on $N=73$), liver ($N=208$), and whole blood ($N=670$) from the GTEx project v8 [6] and micro-dissected kidney tubule ($N=121$) from Doke et al. [21]. GWAS summary statistics for kidney function (measured as eGFRcr, eGFRcys, and BUN; $N=1,004,040; 460,826;$ and $243,031,$ respectively) and kidney damage (ACR; $N=547,361$) were obtained from the Chronic Kidney Disease Genetics (CKDGen) Consortium [16–18, 22].

TWAS of eGFRcr and eGFRcys identified 849 and 416 transcript associations, respectively (Table 1; Fig. 2; Additional file 1: Table S1; Additional file 2: Fig. S1; $P < 3.9 \times 10^{-6}$, "Methods"). Approximately 43% of these associations originated from the whole blood TWAS, reflecting the wider coverage of the transcriptome and better prediction models given the larger sample size for the underlying expression Quantitative Trait Loci (eQTL) analyses. There were 229 shared associations across eGFRcr and eGFRcys, thus representing kidney function rather than effects of creatinine- and cystatin C metabolism. In contrast, the strongest hits identified only for eGFRcr or eGFRcys are known

**Table 1** Number transcripts associated with kidney function and damage ($P < 3.9 \times 10^{-6}$)

| Tissue | Trait | | | |
|---|---|---|---|---|
| | eGFRcr GWAS $N = $ 1,004,040 | eGFRcys GWAS $N = $ 460,826 | BUN GWAS $N = $ 243,031 | ACR GWAS $N = $ 547,361 |
| Kidney cortex # models = 2633, eQTL $N = 73$ | 88 | 55 | 6 | 8 |
| Kidney tubule # models = 1875, eQTL $N = 121$ | 146 | 65 | 16 | 15 |
| Liver # models = 5213, eQTL $N = 208$ | 249 | 113 | 20 | 23 |
| Whole blood # models = 9388, eQTL $N = 670$ | 366 | 183 | 33 | 51 |

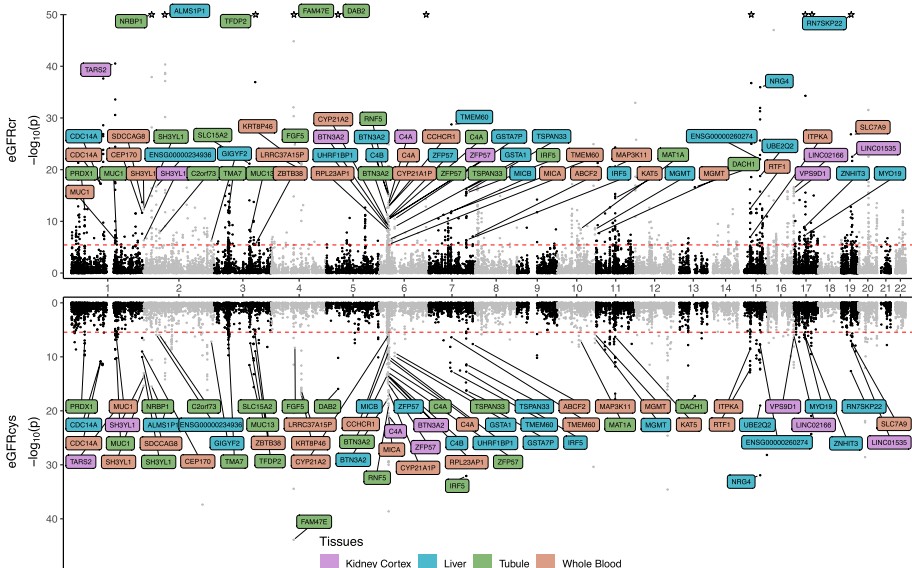

**Fig. 2** TWAS analyses highlight gene expression in kidney-related tissues consistent with genetic determinants of both eGFRcr and eGFRcys. Genes that were significant for the estimated glomerular filtration rate based on creatinine (eGFRcr) and for the estimated glomerular filtration rate based on cystatin (eGFRcys) and that were additionally supported by colocalization analyses of eGFR and expression quantitative trait loci (posterior probability > 0.8) were labeled. The color code indicates the tissue of the TWAS model and $-\log_{10}$(P-values) were capped at 50 (associations indicated by stars). The red lines indicate the Bonferroni adjusted significance threshold ($P < 3.9 \times 10^{-6}$)

to be involved in creatinine and cystatin C production (*GATM* and *CST3*, respectively). The strongest associations among the 229 shared signals were observed for *DAB2* (tubule), *SHROOM3* (liver), *AMLS1P1* (liver), and *NRBP1* (tubule) (Fig. 2; Additional file 1: Table S1).

The TWAS for BUN and ACR, although based on smaller GWAS studies [16, 18], identified 75 BUN and 97 ACR associations (Table 1; Additional file 1: Table S1; Additional file 2: Fig. S2-3; $P < 3.7 \times 10^{-6}$, "Methods").

Next, to minimize the risk of genetic confounding, we performed colocalization analyses for each identified transcript in the TWAS analyses. We identified independent signals of the underlying GWAS summary statistics based on conditional analyses

("Methods"). The median number of independent signals per *cis*-eQTLs was one, and 30 loci displayed multiple signals. The median number of independent signals per GWAS of kidney traits was again one and 343 loci had multiple signals. We observed a colocalization (PP > 0.8; "Methods") of eQTLs and kidney function traits for 288 of the 1437 combinations originating from 214 unique genes (Fig. 2, Additional file 2: Fig. S2-3, Additional file 1: Table S2).

### Proteome-wide association studies (PWAS) of kidney function and damage

We conducted PWAS using models for 1342 circulating proteins constructed in the European ancestry (EA) subpopulation of the Atherosclerosis Risk in Communities (ARIC) study [11]. GWAS summary statistics for kidney function (eGFRcr, eGFRcys, BUN) and damage (ACR) were obtained from the CKD Gen Consortium, as was done in the TWAS analyses [16–18].

PWAS of eGFRcr and eGFRcys identified 69 and 41 associations, respectively (Additional file 1: Table S3; Additional file 2: Fig. S4; $P < 3.7 \times 10^{-5}$, "Methods"). There were 22 shared associations across eGFRcr and eGFRcys, thus likely representing kidney function rather than creatinine- and cystatin C-metabolism-related effects (Fig. 3). The strongest associations among these were observed for IDI2, SNUPN, and INHBC (Additional file 1: Table S3). For signals associated with a single marker alone, the strongest overall associations were observed for CST3 and CST4 for eGFRcys (*P*-values below $1 \times 10^{-320}$ and *z*-score $= -117.8$ and 63.6, respectively), highlighting the contribution of the genes in cystatin C synthesis.

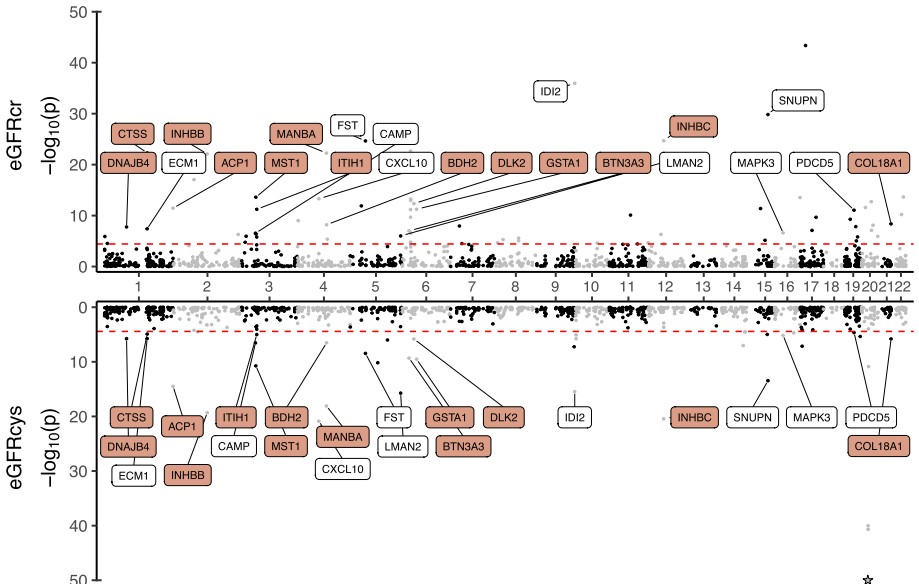

**Fig. 3** PWAS analyses identify circulating proteins consistent with genetic determinants of both eGFRcr and eGFRcys. Proteins that were significant for both the estimated glomerular filtration rate based on creatinine (eGFRcr) and for the estimated glomerular filtration rate based on cystatin (eGFRcys) were labeled. Proteins with additional support through colocalization analyses of eGFR and protein quantitative trait loci (posterior probability > 0.8, "Methods") were highlighted in orange and $-\log_{10}$(*P*-values) were capped at 50 (stronger associations indicated as stars). The red lines indicate the Bonferroni adjusted significance threshold ($P < 3.7 \times 10^{-5}$)

The PWAS for BUN identified ACP1, INHBC, and two aptamers of RSPO3 (SeqId_13094_75 and SeqId_8427_118; Additional file 1: Table S3; Additional file 2: Fig. S5-6; $P < 3.7 \times 10^{-5}$, "Methods"). ACR showed ten significant associations, with CSK exhibiting the strongest ACR association ($P = 1.2 \times 10^{-11}$).

The identified proteins in the PWAS analyses were further studied by conditionally independent colocalization analyses, similar to the procedure performed for the transcripts identified in the TWAS analysis ("Methods"). The median number of independent signals per *cis*-protein quantitative trait locus (pQTL) was three and 100 loci displayed multiple signals, while the kidney function traits again had a median of one and 31 loci with multiple signals. We observed a colocalization (PP > 0.8; "Methods") for 53 of the 124 associations (Fig. 3, Additional file 2: Fig. S5-6, Additional file 1: Table S2 and 3, 30 eGFRcr, 16 eGFRcys, 2 BUN, 5 ACR). For ACP1 and INHBC, two consistent PWAS associations across eGFRcr, eGFRcys, and BUN, we observed colocalizations across all three traits (Fig. 3, Additional file 2: Fig. S4).

### Integration of transcriptome- and proteome-wide association studies using conditional analyses

Combining the TWAS and PWAS results, there were 260 genomic regions encapsulating 398 trait-region associations. Genomic regions were defined by merging all overlapping gene windows (gene start/stop + / − 500 kb) of significant transcripts and proteins across the four traits ("Methods"). Through the iterative nature of these merges, some genomic regions—even outside the HLA region—spanned more than 5 Mb and thus are expected to harbor several independent associations. This procedure ensures that subsequent conditional analyses for each trait (performed separately, between all transcripts and proteins in the region) encompass the most comprehensive set of comparisons. The colocalization analyses supported 196 trait-region associations (PP > 0.8; 110 eGFRcr, 54 eGFRcys, 12 BUN, 20 ACR) within 153 regions (Fig. 4a; Additional file 1: Table S4; labeled as region number 1–153). Using the most significant regional association extracted from the corresponding source GWAS of kidney function and damage, 168 of the trait-region associations reached the standard common-variant GWAS threshold ($P$-value $< 5 \times 10^{-8}$) and 28 were only detected through TWAS/PWAS and did not reach GWAS significance (i.e., GWAS P-value $> 5 \times 10^{-8}$). These novel associations were detected through a combination of power gain and reduced multiple testing burden of the TWAS and PWAS approach, indicating that these approaches can aid in the identification of the underlying mechanisms of complex traits. Furthermore, they may nominate both the molecular mechanisms as well as potentially relevant tissues.

Nearly half of the regions identified in eGFRcys analyses were also supported by eGFRcr analyses (26 of 54), and more than half of the regions in BUN analyses were supported by both eGFRcr and eGFRcys (7 of 13). In contrast, there was less overlap between ACR (a damage marker) and the three filtration markers. Only 3 of 20 ACR regions were supported by at least two of the other traits. Two regions—the *NRBP1* (tubule) and *MUC1* (tubule, whole blood)—were supported by all four traits.

For 58 of the 196 trait-region associations supported by colocalization analysis, significant colocalization was observed for a single tissue and gene, which was prioritized. For the remaining 138 associations, we observed colocalization signals with

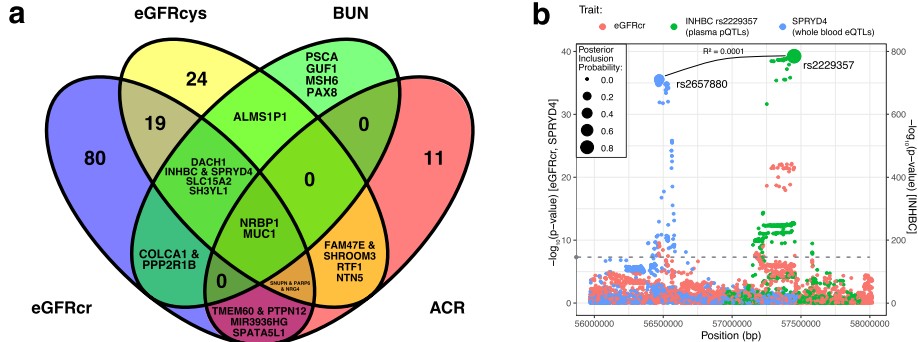

**Fig. 4** Shared and distinct genomic regions underlying the kidney function and damage markers. **a** Venn diagram of the 153 genomic regions identified through TWAS/PWAS and additionally supported by colocalization across eGFRcr, eGFRcys, BUN, and ACR. For intersections with less than five regions, the prioritized genes instead of the number regions were listed. Multiple independent genes pertaining to the same region were separated by an ampersand. **b** Regional association plot for the shared association of the filtration markers represented by eGFRcr that corresponds to independent associations with plasma INHBC protein levels (conditional independent signal) and whole blood *SPRYD4* expression (marginal statistics). The gray dashed line indicates genome-wide significance ($5 \times 10^{-8}$) on the eGFRcr/*SPRYD4* y-axis. INHBC *p*-values are plotted on a separate *y*-axis. Expression quantitative trait loci (QTLs) and protein QTLs were annotated by the posterior inclusion probabilities of SNPs being the driving variant in the region (dot size, "Methods")

multiple tissues or genes. With the goal of prioritizing the model underlying each genomic region or identifying multiple independent associations, we performed conditional analyses. To do this, we first estimated the *cis*-regulated genetic correlation, i.e., the genetically encoded co-regulation within the comparison TWAS and PWAS models ("Methods"). We then used these co-regulation estimates to perform conditional analyses using a regression with summary statistics approach [11, 23]. In this manner, we compare between not only different genes but also different tissues and plasma (Additional file 1: Table S4).

As an example, on chromosome 12, there were several shared associations for eGFRcr, eGFRcys, and BUN, implicating both *INHBC* (colocalizing with plasma protein) and *SPRYD4* (colocalizing with whole blood expression) (Additional file 1: Table S4; region number 103). Plasma levels of INHBC and whole blood *SPRYD4* expression demonstrated negative co-regulation (*cis*-regulated genetic correlation $= -0.04$) and the genetic lead variants of the pQTLs and eQTLs were independent of each other ($R^2 = 0.0001$, Fig. 4b). The conditional analysis of the two co-regulation estimates demonstrated even stronger association signals for both genes, providing support for there being two independent associations in the same region, i.e., INHBC & *SPRYD4* (Fig. 4).

Another illustrative example was seen on chromosome 2, where a shared signal was observed between eGFRcr, eGFRcys, and BUN with *SH3YL1* (colocalizing with gene expression in kidney cortex, tubule, and whole blood) and *ACP1* levels (colocalizing with plasma protein) (Additional file 1: Table S4; region number 19; Fig. 5a). Distinguishing between the transcript and protein signals is complicated by the fact that there are no models of SH3YL1 for circulating proteins nor *ACP1* transcription models available. However, both signals colocalized with the same eGFR peak (PP > 0.8). Conditional analysis showed a markedly strong negative co-regulation of tubule

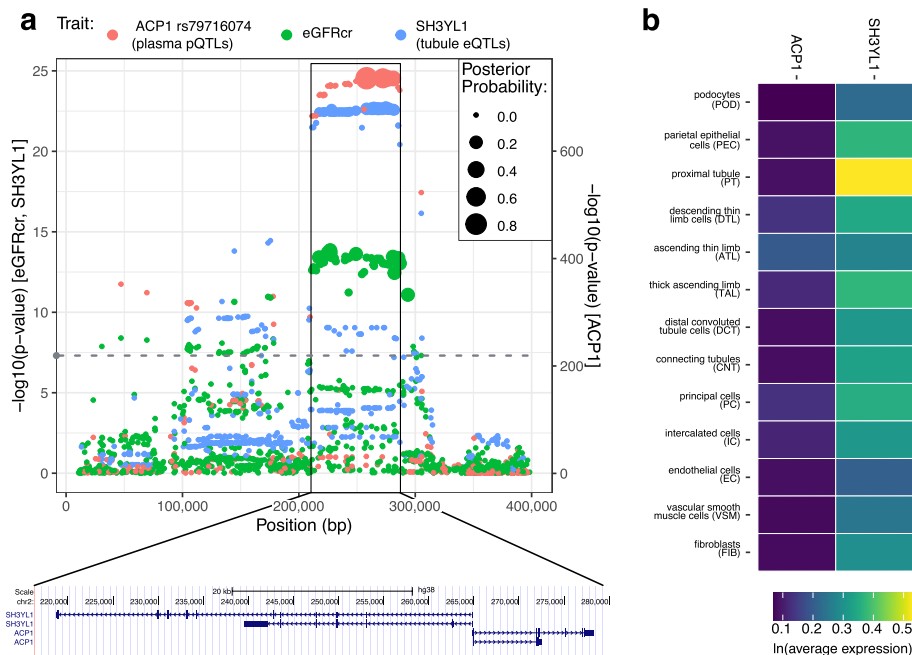

**Fig. 5** Regional association plot of the single signal in the ACP1 / *SH3YL1* region. **a** Regional association plot for the shared association of the filtration markers represented by eGFRcr that identified the same association for plasma ACP1 protein levels (independent signal indexed by rs79716074) and tubule *SH3YL1* expression (marginal statistics). The gray dashed line indicates genome-wide significance ($5 \times 10^{-8}$) on the eGFRcr/*SH3YL1* y-axis. ACP1 p-values are plotted on a separate y-axis. SNPs were annotated by the posterior probabilities of them being the driving variant in the region (dot size, "Methods"). **b** Single-cell RNA sequencing levels across kidney cell types of *SH3YL1* and *ACP1* in the Kidney Precision Medicine Project (KPMP) are displayed. KPMP acute kidney injury samples were excluded

*SH3YL1* and plasma ACP1 (cis-regulated genetic correlation $= -0.92$) thereby pointing to the same underlying association. Tissue-wise, the strongest association was observed for whole blood with a high correlation with kidney tubule *SH3YL1* (cis-regulated genetic correlation $= 0.89$). Subsequently comparing single-cell RNA expression among cell types in the kidney tissue atlas from the Kidney Precision Medicine Project (KPMP) showed a higher expression of *SH3YL1* than *ACP1* with a specifically high expression for proximal tubule (Fig. 5b).

As a final example of using conditional analyses of co-regulation estimates to discern between shared signals, the genomic region on chromosome one near *MUC1* (region number 13) displayed signals for eGFRcr, eGFRcys, BUN, and ACR. Depending on the trait, the association signal was strongest for whole blood or tubule models (cis-regulated genetic correlation $= 0.66$), and for each trait at least one of these associations was additionally supported by colocalization (Additional file 2: Fig. S7). *MUC1* encodes an O-glycosylated protein that helps maintain the protective mucous barriers of epithelial cells. A variable number of tandem repeat (VNTR) region of *MUC1* is associated with the formation of a frameshift, which results in a truncated protein responsible for a rare form of CKD, autosomal dominant tubulo-interstitial kidney disease (ADTKD) [24, 25]. Based on the TWAS models, we connected *MUC1* to a range of kidney function and damage markers in the general population as well (Additional file 2: Fig. S7).

### Higher levels of INHBC and INHBB precede CKD progression

The two strongest PWAS associations supported by colocalization for eGFRcr and eGFRcys were INHBC on chromosome 12 and INHBB on chromosome 2. Both encode members of the TGF-beta (transforming growth factor-beta) superfamily of proteins, specifically subunits of activin complexes. To evaluate whether levels of INHBB and INHBC were associated with CKD progression (ESKD or doubling of serum creatinine), we tested measured INHBB and INHBC protein abundance in the African American Study of Kidney Disease and Hypertension (AASK), an external cohort of CKD patients followed for a median of 8 years and 10 months. Both proteins showed significant associations with CKD progression when adjusting for age, sex, smoking, history of cardiovascular disease, systolic blood pressure, and high-density lipoprotein (HDL) levels (Additional file 1: Tab. S5; "Methods"). Concordant with the inverse relationship of protein abundance and eGFR estimated in the PWAS, hazard ratios in AASK indicated an increased risk for CKD progression with increased protein levels. The association between INHBB and CKD progression was attenuated after adjusting for measured GFR, but the association between INHBC and CKD progression remained significant (HR 1.86 per doubling in INHBC levels, $p = 5.9 \times 10^{-5}$).

### Druggability of identified targets

For the 153 trait-region associations with support from colocalization analyses, we performed annotation of the regions using the open targets database [26]. There were existing drugs that target 29 of the genes, including 14 genes with drugs already approved for use in specific diseases (Additional file 1: Tab. S6). Disease categories of these drugs ranged from diabetes mellitus (*CD86*) to urinary tract infection (*DPEP1*) to COVID-19 (*CASP9*). Of note, *MUC1* (Additional file 2: Fig. S7; Additional file 1: Tab. S1) is targeted by HuHMFG1 [27, 28], SAR-566658 (ClinicalTrials.gov Identifier: NCT02984683), and cantuzumab ravtansine (Drugbank accession number: DB05594). HuHMFGI specifically targets a sequence within the VNTR. However, *MUC1* appears to cause lower levels of BUN (*z*-score in whole blood and tubule TWAS − 10.9 and − 10.6) and thus might be best addressed with an agonist.

### Discussion

Using a cross-omics integrative approach, we generated a genome-wide catalog of potentially causal contributors to CKD. We identified 1561 associations between genetic models of gene expression and proteins with genetic determinants of three markers of kidney filtration (eGFRcr, eGFRcys, BUN) and one of kidney damage (ACR). Through support by colocalization and conditional analyses, we structured these findings into 196 genomic region-trait combinations. For these regions, our results provide insight into gene identification, confirm previous findings in experimental models, identify tissue sites of action, distinguish markers of creatinine or cystatin metabolism from that of GFR, and nominate early biomarkers of CKD progression.

The approach of TWAS and PWAS integration builds upon GWAS by aiding in the identification of genes underlying loci. For example, while *INHBC* was known to be genetically associated with eGFRcr [29] and upregulated in patients with diabetic nephropathy [30], the neighboring association of *SPRYD4* was less well studied, with

only one study finding higher expression in kidney tissue [31]. Our study provides evidence that these represent two independent signals. In contrast, our study helped identify the signals seen with *SH3YL1* and *ACP1* on chromosome 2 as being one and the same. Previous studies have identified associations between SNPs within *SH3YL1* and various traits including blood pressure [32], HDL [33], BMI-adjusted waist circumference [34], as well as eGFRcr [16], BUN [35], and gene expression of *SH3YL1* in various tissues, including the kidney tubule [16]. Our study established associations between *SH3YL1* and eGFRcys and uncovered extensive negative co-regulation of the *SH3YL1* expression and *ACP1* protein levels (*cis*-regulated genetic correlation = −0.97). Because of the greater expression of *SH3YL1* in kidney cells (Fig. 5b) and support from animal models, we suspect that *SH3YL1* underlies the causal association. Specifically, murine knock-out models of *Sh3yl1* show reduced inflammatory response, tubular apoptosis, renal failure, and mortality after endotoxin exposure [36].

The validity of our approach is strengthened by that fact that several of our findings have orthogonal support from existing animal models. We identified *MANBA* in whole blood and tubule TWAS as well as plasma PWAS for eGFRcr and eGFRcys (region number 51; Fig. 3; Additional file 1: Tab. S4). *MANBA* was recently studied in a murine knock-out model which linked the gene to differential kidney fibrosis after toxic injury induced by cisplatin or folic acid [37]. Similarly, *DACH1*, which was associated with eGFRcr, eGFRcys, and BUN in our study, was previously studed in murine models that demonstrated tubule-specific Dach1 deletion caused more severe renal fibrosis after kidney injury [21].

The integration of TWAS and PWAS can help nominate specific tissues underlying causal associations. *NRBP1* is a known eGFRcr GWAS locus that was previously supported by colocalization with methylation QTLs [38]. Interestingly, nearby CpG methylation sites were associated with eGFRcr and ACR in epigenome-wide association studies (eGFRcr: cg11111225; 514 kb upstream; *P*-value = $2.2 \times 10^{-6}$; ACR: cg23635560; 177 kb upstream; *P*-value = $1.2 \times 10^{-7}$) [39]. Our study adds to this work by confirming associations between *NRBP1* and all of the studied kidney traits, as well as nominating kidney tubule expression as the possible site of action. Consistent with this finding is that knock-down of *NRBP1* in mice resulted in upregulation of *ABCG2*, a urate transporter in the proximal tubule, and that overexpression of *NRBP1* resulted in downregulation of *SLC22A12* and *SLC2A9*, two organic anion transporters found in epithelial cells in the proximal tubule [40].

The combined study of different kidney function biomarkers can isolate genes involved in determining kidney function from those involved in creatinine or cystatin C metabolism. Recently, a study used massively parallel reporter assays to functionally evaluate genetic variants for eQTLs and 114 human traits [41]. They found six independent eQTLs in *SPATA5L1* colocalizing with GWAS summary statistics for CKD (defined using eGFRcr). Another study also identified *SPATA5L1* as one of the strongest eGFRcr GWAS loci and reported a positive colocalization with *SPATA5L1* expression [42]. Consistent with this finding, one of the strongest TWAS associations in our study was observed for *SPATA5L1* and *GATM*, a neighboring gene that was implicated in some of the earliest eGFRcr-based GWAS [43]. However, *GATM* is an essential enzyme in creatine biosynthesis, catalyzing the transfer of a guanido group to the immediate precursor

of creatine. We observed no association of *SPATA5L1* and *GATM* with eGFRcys or BUN, consistent with a role in creatine production but not kidney function per se. In contrast, we found 229 TWAS associations, 22 PWAS associations, and 26 colocalization-supported genomic regions that were shared across eGFR measures and thus likely reflect kidney filtration rather than solely creatinine and cystatin C metabolism.

The genome-wide catalog can nominate new biomarkers of disease progression. Motivated by the two strongest PWAS associations shared for eGFRcr and eGFRcys and supported by colocalization, as well as the fact that the components are subunits of the same protein complex that is encoded on different chromosomes, we performed follow-up analyses in the AASK study of CKD patients for INHBB and INHBC. We found an increased risk for CKD progression per higher protein abundance, consistent with the direction suggested by PWAS. Even after comprehensive adjustment, including for measured GFR, two-fold higher INHBC conferred an 86% increased risk for CKD progression. This observation is also supported in animal models. Higher INHBB was found in a polycystic kidney disease mouse model compared to controls [44], and higher INHBC expression was found in a diabetic nephropathy rat model [45]. Further, inhibition and overexpression models of *INHBB* regulated renal fibrosis [46].

Finally, this study extends support for causal associations across multiple tissues and kidney biomarkers. The association between SNPs in *MUC1* and ACR was previously identified [18]. We found parallel associations with eGFRcr, eGFRcys, and particularly BUN, augmenting support for a role of *MUC1* in kidney disease. Interestingly, these common SNPs are distinct from the known ADTKD-associated frameshift in *MUC1*.

Our work confirms results from previous studies of genetic determinants of kidney function. Zheng et al.screened proteomic genetic instruments for effects on 223 traits, including eGFRcys, identifying one association supported by colocalization (CST3), and two that were not supported by colocalization (CST4 and CST5) [13]. We replicated these observations and confirmed no association between CST3 and the other markers of kidney filtration, suggesting that CST3 (cystatin) is a marker but not causal for kidney function itself. In a separate study, Matias-Garcia et al. [4] used MR methods to implicate three proteins with a causal effect on eGFRcr: MIA, CA3, and CST6. We confirmed the effects of MIA and CA3 on eGFRcr, providing additional support by colocalization studies (CST6 was not available as a PWAS model). Hellwege et al. [42] identified 45 significant genes using a kidney-specific TWAS for eGFRcr, 19 of which were supported by colocalization. Our study used a non-overlapping source for GWAS summary statistics (theirs used a smaller trans-ethnic GWAS) and replicated 15 of the 16 available in our TWAS screen. Finally, Doke et al. [21] identified 51 significant genes in a tubule-specific TWAS using an older GWAS ($n = 765{,}348$). Using an updated GWAS of kidney function ($N > 1$ million) [17], we confirmed previous findings and increased the number of significant tubule TWAS associations for eGFRcr to 146. Overall, we identified 234 significant associations in the kidney cortex or tubule.

Some limitations warrant mentioning. First, TWAS and PWAS associations displayed genomic control lambdas > 1 (Additional file 2: Fig. S1 and S4). These inflations are expected and relate to the well-powered source GWAS statistics (genomic inflation factor lambda: eGFRcr 1.24, eGFRcys 1.17, BUN 1.15, ACR 1.19), which reflect the polygenic nature of the traits. The meta-analyses corrected for inflation on the individual

study level. A priori, we would expect that PWAS and TWAS would have even larger lambdas given the stronger demonstration of polygenicity. Reassuringly, the PWAS models displayed no inflation when tested based on simulated null data [11]. Second, the tissue-specific TWAS and PWAS models have different coverage of the genome. Due to varying sample sizes of tissue-specific mRNA, the number of prediction models ranged from 1870 (tubule) to 9388 (whole blood), and the number of protein models was 1342. Third, we required colocalization for identification of top signals, which may have resulted in false negatives. Fourth, regions of extreme linkage disequilibrium and/or very strong associations can lead to spurious results of the conditional GCTA analysis detecting many independent associations. However, many of the observations with multiple independent associations are biologically plausible. For the 24 associations involving a GWAS signal with more than seven independent associations, all except one (plasma MICB—eGFRcr) were either supported by other kidney traits or showed no colocalization overall, hence limiting the risk of false positives. Fifth, we used GWAS summary statistics from a European-American population, limiting generalizability. We chose this data source to better reflect the populations underlying the TWAS and PWAS weights, thereby avoiding potential bias due to differences in allele frequencies. Additional studies are needed to combine high-quality GWAS summary statistics from more diverse populations [11]. Lastly, the sample sizes of the source GWAS ranged from 243,031 (BUN) to one million (eGFRcr), which limits the comparison across filtration markers due to different power to detect signals. We focused on the overlap of eGFRcr and eGFRcys hits instead of that of eGFRcr, eGFRcys, and BUN due to the significantly lower sample size of the BUN GWAS as well as the idea that eGFRcr and eGFRcys are estimating the same latent trait (kidney function).

Strengths of this study include the comprehensive scale of the screened models, covering 12,893 genes and 1342 proteins across five kidney function-related tissues (liver, kidney cortex, plasma, tubule, and whole blood). As the TWAS and PWAS models do not require GWAS trait associations in the same dataset, we were able to leverage summary statistics from consortia of much larger sample size, maximizing the power to detect associations. By anchoring to cis-based genetic models and performing additional colocalization analyses, we reduced the risk of confounding and reverse causation. We integrated tissue-specific TWAS and plasma PWAS findings as well as co-regulated neighboring genes through conditional models [11]. Furthermore, we conducted the follow-up analyses for *INHBC* and *INHBB* in the AASK study that represents both a switch to a cohort with prevalent CKD and a cohort of African Ancestry. This strengthens the generalizability of our findings for these targets. Finally, we isolated potential markers of true kidney function rather than biomarker-specific effects using the combination of eGFRcr, eGFRcys, and BUN as kidney function markers and ACR as a kidney damage marker.

## Conclusions

In summary, this study generates a catalog of putatively causal target genes, tissues, and proteins relevant to kidney function and damage and constitutes a comprehensive resource to guide follow-up studies in physiology, basic science, and clinical medicine.

## Methods

### Summary statistics

Summary statistics from meta-analyses of Genome-wide Association Studies (GWAS) of European ancestry were obtained from the CKDGen consortium (Stanzick et al.: eGFRcr & eGFRcys; Wuttke et al.: BUN; Teumer et al.: ACR). [16–18]

### Transcriptome-wide association studies (TWAS)

We performed TWAS following the FUSION workflow based on weights for the kidney function relevant tissues from GTEx v8 (kidney cortex, liver, and whole blood, http://gusevlab.org/projects/fusion/) [6] and kidney tubule from Doke et al. [21] Prediction models were based on elastic net modelling and combined with the accompanying European ancestry 1000 Genomes Project linkage disequilibrium (LD) reference (https://data.broadinstitute.org/alkesgroup/FUSION/LDREF.tar.bz2). Multiple testing was accounted for by a Bonferroni adjustment for the number of genes modelled across tissues that overlapped at least one of the GWAS summary statistics datasets (*P*-value $< 3.9 \times 10^{-6} = 0.05$ / 12,893 unique genes). Manhattan and Miami plots were created based on the miamiplot R package (https://github.com/julie dwhite/miamiplot).

### Proteome-wide association studies (PWAS)

For PWAS, we applied the same FUSION workflow based on elastic net modelling as for TWAS. Weights were based on the European ancestry subpopulation of the ARIC and combined with the accompanying European ancestry in-sample LD reference (http://nilanjanchatterjeelab.org/pwas/) [11]. Multiple testing was accounted for by a Bonferroni adjustment for the number of aptamers modelled that overlapped at least one of the GWAS summary statistics datasets (*P*-value $< 3.7 \times 10^{-5} = 0.05$ / 1342 unique aptamers).

### Conditional independent colocalization analysis

For each of the significant associations, we performed colocalization analyses of independent signals [13, 15, 47]. Hence, we extracted the eQTL / pQTL and kidney function / damage summary statistics for the underlying genes with a 250-kb flanking region [48] and identified independent associations based on approximate conditional analyses by the GCTA COJO-Slct algorithm, a step-wise-forward-selection approach (*P* conditional $< 5 \times 10^{-8}$), using a collinearity cut-off of 0.1 [47]. The matching European ancestry LD reference from the ARIC study was used [11]. If multiple independent SNPs were identified, summary statistics for each SNP were computed by conditioning on all other independent SNPs in the gene region using the GCTA COJO-Cond algorithm (collinearity cut-off = 0.1, same LD reference) [47]. Finally, approximate Bayes factors were estimated and used to compute posterior inclusion probabilities (PIP) of SNPs being the driving variant in the region [49]. Colocalization analyses were conducted based on the SNP-wise PIPs for all pairwise combinations of independent eQTL / pQTL associations against the independent kidney function / damage GWAS associations [15, 49]. For this, an adapted version of the

Giambartolomei colocalization method as implemented in the "coloc.fast" function (https://github.com/tobyjohnson/gtx) with default parameters and prior definitions was used. If one or no independent genome-wide significant association was detected, marginal summary statistics were used. Colocalizations were reported if the PP of one common causal variant (H4 / p12) for a combination of independent associations reached 0.8 [15]. eQTL and pQTL data was available for 1472 of the 1477 significant TWAS / PWAS associations [11, 21, 50].

### Genomic region definition and conditional analyses

To comprehensively select all TWAS and PWAS associations for comparative conditional analyses, we defined genomic regions by combining overlapping association signals. All significant association from TWAS / PWAS across the four traits were recursively merged into genomic regions if the gene region $+/-$ 500 kb overlapped. The MHC region on chromosome 6 was considered as one region spanning from 25.5 to 34 Mb. The resulting 260 genomic regions corresponded to 398 region-trait combinations.

For each region-trait combination that was supported by at least one colocalization (PP > 0.8), we performed comparative conditional analyses to prioritize genes and tissues. All associations in the region that were supported by colocalization and the ones with the minimum TWAS / PWAS *p*-value per tissue were included in the conditional analyses. We imputed the respective transcripts and proteins for individuals from Phase-3 1000 Genome Project (1000Genome) to estimate the *cis*-regulated genetic Pearson's correlation coefficients [51]. Similar to the methodological foundation of GCTA COJO-Cond using *z*-scores and the *cis*-regulated genetic correlation of the associations [47], we estimated the pairwise conditional associations as described earlier [11]. This allowed us to study whether one transcript / protein in a certain tissue explains the association signal of another.

### CKD progression associations of INHBB and INHBC in the AASK study

The AASK study was a trial of African Americans aged 18–70 years with hypertensive chronic kidney disease (mGFR 20–65 ml/min per 1.73 $m^2$) [52]. All 705 participants with available proteomic profiling at baseline in the trial phase were included in our analysis. Protein levels were $log_2$-transformed. All patients were treated with antihypertension medication and had no diabetes at baseline. Cox proportional hazards models were used to relate each protein to risk of CKD progression, defined as time to ESKD or doubling of serum creatinine. Analyses were initially adjusted for age, sex, systolic blood pressure, history of cardiovascular disease, smoking (current/past/never), and HDL levels; subsequent models had additional adjustment for mGFR. Statistical significance was determined using a Bonferroni correction (*P*-value < 0.05/2).

### Single-nucleus RNA sequencing

Single-nucleus RNA sequencing data for *SHY3L1* and *ACP1* was extracted through the NephGen scExplorer (https://nephgen.imbi.uni-freiburg.de) from the Kidney Precision Medicine Project – Kidney Tissue Atlas (https://atlas.kpmp.org/) [53]. Cells from patients with acute kidney injury were removed and sub cell types were combined by cell count weighted averages. We have complied with all ethical regulations related to

this study. Human samples collected as part of the Kidney Precision Medicine Project (KPMP) consortium were approved as exempted by the University of Washington Institutional Review Board, and informed consent was obtained for the use of data and samples [53].

## Druggability of targets

Known drugs and tractability data were annotated based on the Open Targets Platform (accessed 3/17/22; Additional file 1: Tab. S6) [26]. Approval status was indicated as true if any approved drug exists for a given gene.

## Supplementary Information

---

**Additional file 1.** Supplementary tables listing the results for TWAS, colocalization analyses, PWAS, conditional analysis, prospective analysis and the pharmacological annotation of identified targets.

**Additional file 2.** Supplementary Figures including the QQ-plots, Manhattan plots and the MUC1 TWAS association.

**Additional file 3.** Review history

---

### Acknowledgements
The authors thank the staff and participants of the AASK study for their important contributions. The opinions presented do not necessarily represent those of the NIDDK, the NIH, the Department of Health and Human Services, or the US Government. The interpretation and reporting of these data are the responsibility of the authors and in no way should be seen as an official policy or interpretation of the US Government. We thank Burulca Göcmen and Dr. Anna Köttgen for their support and discussions through CRC 1453 NephGen. We thank Dr. Junghyun Jung, Dr. Nicholas Mancuso, and Dr. Alexander Gusev for providing early access to the TWAS models built with GTEx v8 data.

### Review history
The review history is available as Additional file 3.

### Peer review information

### Authors' contributions
Research idea and study design: P.S., M.E.G.;
Data acquisition: M.E.G.;
Data analysis/interpretation: P.S., A.L.S., M.E.G.;
Supervision or mentorship: P.S., M.E.G.
Each author contributed important intellectual content during manuscript drafting or revision and agrees to be personally accountable for the individual's own contributions and to ensure that questions pertaining to the accuracy or integrity of any portion of the work.

### Funding
The work of P.S. was supported by the German Research Foundation (DFG) Project-ID 431984000—CRC 1453 NephGen, Project-ID 192904750 – CRC 992 Medical Epigenetics, Project-ID 523737608 (SCHL 2292/2–1), and the EQUIP Program for Medical Scientists, Faculty of Medicine, University of Freiburg. The work of M.E.G. was funded by NIDDK: R01 DK124399, NHLBI: K24 HL155861, R01DK108803. The work of J.C. was funded by NIDDK: R01 DK124399. The work of J.Z., and N.C. was supported by NHGRI: R01 HG010480-01.

### Availability of data and materials
Source GWAS summary statistic datasets are available from the CKDGen Consortium (https://ckdgen.imbi.uni-freiburg.de) [16-18] and the scExplorer (https://nephgen.imbi.uni-freiburg.de). Pre-existing data access policies for the AASK cohort study specifies that research data requests can be submitted to the steering committee; these will be promptly reviewed for confidentiality or intellectual property restrictions and will not unreasonably be refused. Please refer to the data sharing policies on https://repository.niddk.nih.gov/studies/aask-trial/.

## Declarations

### Ethics approval and consent to participate
The African American Study of Kidney Disease and Hypertension clinical protocol was approved by the Institutional Review Board (IRB) of each participating institution (IRB00116062), and each patient provided informed consent.

### Consent for publication
All authors have approved the manuscript and give their consent for submission and publication.

## Competing interests
The authors declare that they have no competing interests.

## Author details
[1]Department of Epidemiology, Johns Hopkins Bloomberg School of Public Health, Baltimore, MD, USA. [2]Department of Biostatistics, Johns Hopkins Bloomberg School of Public Health, Baltimore, MD, USA. [3]Department of Medicine and Genetics, Perelman School of Medicine, University of Pennsylvania, Philadelphia, PA, USA. [4]Welch Center for Prevention Epidemiology and Clinical Research, Johns Hopkins University, Baltimore, MD, USA. [5]Division of Precision Medicine, New York University Grossman School of Medicine, New York, NY, USA. [6]Nephrology Division and Endocrine Unit, Massachusetts General Hospital, Boston, MA, USA. [7]McKusick-Nathans Institute, Department of Genetic Medicine, Johns Hopkins University School of Medicine, Baltimore, MD, USA. [8]Epidemiology, Human Genetics and Environmental Sciences, School of Public Health, University of Texas Health Science Center at Houston, Houston, TX, USA. [9]Human Genome Sequencing Center, Baylor College of Medicine, Houston, TX, USA. [10]Department of Medicine, Johns Hopkins University School of Medicine, Baltimore, MD, USA. [11]Department of Physiology, Johns Hopkins University School of Medicine, Baltimore, MD, USA.

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

## 