## [**Additional file 3.** Review history · Genome Biology]

Review History

First round of review

Reviewer 1

Are you able to assess all statistics in the manuscript, including the appropriateness of statistical tests used? Yes, and I have assessed the statistics in my report.

Comments to author:

This manuscript describes a TWAS and colocalization analysis of previously published GWAS studies for several parameters of renal health. The authors extend the inferences by applying a pharmacological overlay to assess potential for repurposing existing treatments for other conditions. The premise is logical and scientifically sound, although not entirely novel. The methods are standard and

Hellwege et al. (PMID: 31451708) conducted a similar analysis with older methods and different GWAS resources, and so a comparison of those results with these should be included in the Discussion and Results. Other prior papers have also evaluated colocalization with renal function parameters, but this is the most direct comparison.

Additionally, another sub-optimal aspect of the study design is the lack of non-European ancestry GWAS in the analysis. GWAS in general have not achieved adequate diversity in large-scale studies, but there have been some trans-ancestry studies of renal function that are worthy of note. Excluding those studies weakens the quality of this work considerably in the current context in the field. At the least, this should be acknowledged in the Discussion as a weakness.

In general, the writing is very long and could be consolidated to benefit readers who desire a concise summary of findings. There are some parts of the results that read like Discussion (the MUC1 story in Results, the INHBC and INHBB section, etc.), and both the Introduction and Discussion could be shortened quite a bit.

Page 18, line 257: What is a statistical abstraction? Are there methods or results to describe for this?

Figure 6 is complex and is perhaps not very effective. If there is an effort to simplify the presentation this could be eliminated.

The primary results from the study are presented in ST1. There is no table footnote explaining the column names. The directions of effect are only presented via the TWAS Z statistic, which conflates the magnitude of the effect with the evidence for the effect not being 0. Can the actual effect of changing expression (per SD or similar) be derived from this data? This is available from the S-PrediXcan method, but not sure about FUSION. Also it should be stated or specified that all effects are with regard to increasing gene expression by some amount.

Were TWAS/PWAS directions of effect and drug mode of action taken into account in the drug screen? If not, then a drug might either worsen or improve the outcome. For instance, if the association is between a gene/protein where increasing expression is associated with decreasing eGFR, and the drug is an agonist for that gene/protein, then that drug might not be a good idea to use as treatment for someone in renal decline. Similar logic for a gene/protein where increasing levels are associated with improved renal parameters and the drug inhibits that gene/protein.

GC lambda stats are not really an appropriate way to evaluate bias in very large GWAS of highly complex traits. LD score intercepts are a more accepted means of evaluating that, and even they can suffer from extreme polygenicity in high-powered studies.

The authors make some statements about a comprehensive interpretation of associations, but this seems an exaggeration. There are no attempts to evaluate biological pathways or broader biomedical context using phewas or FUMA or similar.

Reviewer 2

Are you able to assess all statistics in the manuscript, including the appropriateness of statistical tests used? No, I do not feel adequately qualified to assess the statistics.

Comments to author:

I can comment on the apparent significance of the findings and conclusion, with the integrated TWAS, PWAS and GWAS bringing new insights which should be of value in further exploratory studies.

Reviewer 3

Are you able to assess all statistics in the manuscript, including the appropriateness of statistical tests used? Yes, and I have assessed the statistics in my report.

Comments to author:

Schlosser et al conducted an integrated analysis of TWAS + PWAS on kidney function and damage traits, followed by conditional analysis (comparison of TWAS/PWAS findings) and observational analysis. The state-of-the-arts methods have been applied. The manuscript is well written, e.g. explain why liver have been selected as a kidney related tissue. The whole story prioritised a list of genes that with multiple layers of evidence on kidney function/damage, which could be considered as drug targets for future trials. In general, these group of experts provided a comprehensive story here. I have some comments, which hopefully will make the manuscript in a better shape.

Major comments:

1. Selection of genes for kidney function. Considering both eGFRcr and eGFRcys signals is a nice way of selecting genes associated with kidney function. One suggestion here. Since the

sample size of eGFR_{cr} GWAS is twice as eGFR_{cys} GWAS. It will be possible that some TWAS signals showed strong effects on eGFR_{cr} but only showed marginal effects on eGFR_{cys} due to power issue. This may help pick some of the real signals back.

One related question, does TWAS results of BUN been treated as a filtering step for kidney function associated genes? It is not directly clear what is the value of including BUN if it has not been used as a selection criteria.

2. Power of PWAS and TWAS, pQTL data were measured in more samples. However, when comparing top TWAS and PWAS signals, why the number of PWAS signals were much less than TWAS? Is this just an issue of number of proteins been tested?

3. the percentage of TWAS/PWAS signals with coloc evidence is a bit lower than I expected. Is this a kidney function specific thing?

4. Conditional analysis, I think this section is the most value-added section in this manuscript. I roughly understand that this approach helps to identify distinct TWAS/PWAS signals in a region (nice demonstration in Figure 4). However, it is not easy to follow some of the details by reading the results and methods. Few questions:

a. The conditional analysis was conducted between TWAS and PWAS signals for one kidney trait? Or across multiple kidney traits. The cis-regulated genetic correlation is the r_g between transcript and kidney function or r_g between transcript and protein? Does the genetic correlation help for filtering out signals that need to do conditional analysis?

b. 5MB region this is a quite wide range. will it be possible to just use centralised eQTL and pQTL + 500kb each side to select an overlapped region for both eQTL and pQTL in the same genomic region?

5. Linkage between TWAS/PWAS and observational analysis

a. Proteins vs CKD progression, a fair comparison will be comparing PWAS of proteins vs CKD progression with observational evidence. Here, we are just comparing PWAS signal of kidney function/damage in one time point.

b. It is a nice idea to test association of proteins on CKD progression, but why in AASK rather than in ARIC? Since the evidence identified in AASK may reflect ancestry specific effects.

Minor comments:

1. Abstract:

A proportion of the abstract was used to explain the alignment of PWAS/TWAS findings and literature evidence. This give me a wrong impression that the PWAS/TWAS findings were followed by animal models etc. It is not directly clear what is the value added to include all these rather than using more efforts to explain the observational analysis and conditional analysis been actually conducted in this study.

In addition, for the term “several genes”, better to put exact number here.

The term “independent association” is not easy to follow until read over the manuscript. Better to use another term, e.g. independent gene-trait and protein-trait associations in the same genomic regions or so.

2. Line 98-99, cis variants can be pleiotropic too. Better to make it clear here.
3. Line 129, list how many genes had been tested in the TWAS?
4. Figure 2 and 3, no colour and gene label for eGFRcys?
5. Line 156, positive colocalization. Colocalization analysis only provides probability rather than an effect size with direction of effect. It will be better to just say strong coloc evidence rather than positive.

Dear Dr. Pang and dear Reviewers,

Thank you for the thorough evaluation of our manuscript, and for your thoughtful and constructive comments. We have carefully considered them, performed new analyses, and responded to each comment in the point-by-point response, below.

We believe that your comments have helped us to significantly improve our manuscript. We hope that the implemented changes meet your requests, and that we have answered all open questions.

Sincerely,

Pascal Schlosser, and Morgan E. Grams, on behalf of all authors

Point-by-point response

Reviewer #1:

This manuscript describes a TWAS and colocalization analysis of previously published GWAS studies for several parameters of renal health. The authors extend the inferences by applying a pharmacological overlay to assess potential for repurposing existing treatments for other conditions. The premise is logical and scientifically sound, although not entirely novel. The methods are standard and Hellwege et al. (PMID: 31451708) conducted a similar analysis with older methods and different GWAS resources, and so a comparison of those results with these should be included in the Discussion and Results. Other prior papers have also evaluated colocalization with renal function parameters, but this is the most direct comparison.

Response: Many thanks for the thorough review and positive assessment of our work. We also view our study as complementary to the excellent work by Hellwege and colleagues,¹ which included a S-PrediXcan TWAS. We now include a comparison with this and other previous TWAS/PWAS of kidney function¹⁻⁴ (lines 353-61). It warrants mentioning that while Hellwege used a different TWAS methodology, eQTL tissue, and underlying eGFRcr GWAS source (non-overlapping sample), we did replicate 15 of their 16 TWAS hits with colocalization that were available in our analysis, which we believe validates the overall approach. We also built upon previous work by identifying 169 eGFRcr TWAS hits with colocalization. Other ways in which we expanded upon the Hellwege study were by performing TWAS on kidney cortex,

tubule, liver, and whole blood TWAS (Hellwege et al used kidney tissue only) as well as by performing plasma PWAS; and by analyzing eGFRcys, BUN, and ACR in addition to eGFRcr.

Additionally, another sub-optimal aspect of the study design is the lack of non-European ancestry GWAS in the analysis. GWAS in general have not achieved adequate diversity in large-scale studies, but there have been some trans-ancestry studies of renal function that are worthy of note. Excluding those studies weakens the quality of this work considerably in the current context in the field. At the least, this should be acknowledged in the Discussion as a weakness.

Response: We fully agree with the reviewer that lack of diversity is a huge limitation in current genetic studies of kidney function and many other traits. Non-European ancestries are severely underrepresented. This recognition has spurred some ongoing large-scale efforts such as the Million Veterans Program⁵ and the All of Us Research Program⁶ that seek to improve the representation of minority populations. In our manuscript, we chose to restrict the kidney function GWAS summary statistics to European ancestry (instead of trans-ethnic) to better reflect the populations underlying the TWAS/PWAS FUSION weights; this decision was made in order to avoid potential bias due to differences in allele frequencies. Although we did develop weights specific to African American ancestry for PWAS, the publicly available summary statistics for GWAS of eGFR are limited.⁷⁻⁹ In the publicly available Pattaro summary statistics, the effect sizes and standard errors have limited precision (4 digits), small sample size (16,474 compared to N>1 million for EA), and by today's standard patchy imputation (only a fraction of the imputed genotypes used in the generation of PWAS weights are available due to the use of an older panel in Pattaro *et al.*). Were we to use these two sources (Zheng's African Ancestry PWAS weights and Pattaro's African Ancestry GWAS), we could only test 13 of the 1385 proteins represented in the PWAS weights. We believe subsequent efforts with larger data sources of GWAS summary statistics and modern imputation would enable an adequate analysis of non-EA ancestries using the concept of TWAS/PWAS and should be undertaken. We included this in the discussion/limitations section (lines 379-83).

In general, the writing is very long and could be consolidated to benefit readers who desire a concise summary of findings. There are some parts of the results that read like Discussion (the MUC1 story in Results, the INHBC and INHBB section, etc.), and both the Introduction and Discussion could be shortened quite a bit.

Page 18, line 257: What is a statistical abstraction? Are there methods or results to describe for this?

Figure 6 is complex and is perhaps not very effective. If there is an effort to simplify the presentation this could be eliminated.

Response: Thank you for your feedback. We revised the Introduction and Discussion for a more concise presentation while simultaneously addressing the wide audience of *Genome Biology*. Furthermore, we moved **Figure 6** to the Supplement (now **Supplementary Figure 7**), shortened the MUC1 section (which eliminated the sentence on page 18), and moved parts of the MUC1 section to the Discussion.

The primary results from the study are presented in ST1. There is no table footnote explaining the column names. The directions of effect are only presented via the TWAS Z statistic, which conflates the magnitude of the effect with the evidence for the effect not being 0. Can the actual effect of changing expression (per SD or similar) be derived from this data? This is available from the S-PrediXcan method, but not sure about FUSION. Also it should be stated or specified that all effects are with regard to increasing gene expression by some amount.

Response: The original FUSION pipeline does not include the estimation of the effect sizes encoded in the genetically regulated part of the gene expression. We extended the concept and now computed effect estimates based on the transcript / protein prediction models. We revised the **Supplementary Table 1, 3, and 4** such that column names are self-contained, and added new columns representing the effect direction, effect estimates, standard errors and confidence intervals in **Supplementary Table 1 and 3**.

Were TWAS/PWAS directions of effect and drug mode of action taken into account in the drug screen? If not, then a drug might either worsen or improve the outcome. For instance, if the association is between a gene/protein where increasing expression is associated with decreasing eGFR, and the drug is an agonist for that gene/protein, then that drug might not be a good idea to use as treatment for someone in renal decline. Similar logic for a gene/protein where increasing levels are associated with improved renal parameters and the drug inhibits that gene/protein.

Response: Thank you for raising this important issue. In addition to the *MUC1* targeting drugs discussed in the manuscript, we have now reviewed the pharmacological annotations of the identified TWAS/PWAS targets and added the mechanisms of action in **Supplementary Table 6**.

GC lambda stats are not really an appropriate way to evaluate bias in very large GWAS of highly complex traits. LD score intercepts are a more accepted means of evaluating that, and even they can suffer from extreme polygenicity in high-powered studies.

Response: We agree with reviewer. In the section mentioning GC lambdas, our intent was to convey that the observed inflation is likely due to polygenicity instead of bias. We fully agree that LD score regressions as introduced by Bulik-Sullivan, et al. ¹⁰ is an excellent approach to differentiate inflation due to bias and polygenicity in GWAS. However, how to transfer this to the TWAS / PWAS framework and use the full co-regulation matrix instead of an LD matrix requires careful methods development. We revised the discussion accordingly (lines 363-9).

The authors make some statements about a comprehensive interpretation of associations, but this seems an exaggeration. There are no attempts to evaluate biological pathways or broader biomedical context using phewas or FUMA or similar.

Response: We apologies for the misplaced statement which we have removed (e.g., lines 391-8).

References:

1. Hellwege, J.N. *et al.* Mapping eGFR loci to the renal transcriptome and phenome in the VA Million Veteran Program. *Nat Commun* **10**, 3842 (2019).
2. Zheng, J. *et al.* Phenome-wide Mendelian randomization mapping the influence of the plasma proteome on complex diseases. *Nat Genet* **52**, 1122-1131 (2020).
3. Matias-Garcia, P. *et al.* Plasma Proteomics of Renal Function: A Trans-ethnic Meta-analysis and Mendelian Randomization Study. *J Am Soc Nephrol* (2021).
4. Doke, T. *et al.* Transcriptome-wide association analysis identifies DACH1 as a kidney disease risk gene that contributes to fibrosis. *J Clin Invest* **131**(2021).
5. Gaziano, J.M. *et al.* Million Veteran Program: A mega-biobank to study genetic influences on health and disease. *J Clin Epidemiol* **70**, 214-23 (2016).
6. The “All of Us” Research Program. *New England Journal of Medicine* **381**, 668-676 (2019).
7. Zhang, J. *et al.* Plasma proteome analyses in individuals of European and African ancestry identify cis-pQTLs and models for proteome-wide association studies. *Nat Genet*, 2021.03.15.435533 (2022).
8. Pattaro, C. *et al.* Genetic associations at 53 loci highlight cell types and biological pathways relevant for kidney function. *Nat Commun* **7**, 10023 (2016).
9. Fatumo, S. *et al.* Discovery and fine-mapping of kidney function loci in first genome-wide association study in Africans. *Hum Mol Genet* **30**, 1559-1568 (2021).
10. Bulik-Sullivan, B.K. *et al.* LD Score regression distinguishes confounding from polygenicity in genome-wide association studies. *Nat Genet* **47**, 291-5 (2015).

Dear Dr. Pang and dear Reviewer 3,

Thank you for the thorough evaluation of our manuscript that reached us shortly after the formal decision including comments by Reviewer 1 and 2 (see separate file). We have carefully considered your thoughtful and constructive comments, performed new analyses, and responded to each comment in the point-by-point response, below.

We believe that your comments have helped us to significantly improve our manuscript. We hope that the implemented changes meet your requests, and that we have answered all open questions.

Sincerely,

Pascal Schlosser, and Morgan E. Grams, on behalf of all authors

Point-by-point response

Reviewer #3:

Schlosser et al conducted an integrated analysis of TWAS + PWAS on kidney function and damage traits, followed by conditional analysis (comparison of TWAS/PWAS findings) and observational analysis. The state-of-the-arts methods have been applied. The manuscript is well written, e.g. explain why liver have been selected as a kidney related tissue. The whole story prioritised a list of genes that with multiple layers of evidence on kidney function/damage, which could be considered as drug targets for future trials. In general, these group of experts provided a comprehensive story here. I have some comments, which hopefully will make the manuscript in a better shape.

Response: We would like to thank the Reviewer for the positive feedback and the thoughtful suggestions, which we have addressed as outlined below.

Major comments:

1. Selection of genes for kidney function. Considering both eGFRcr and eGFRcys signals is a nice way of selecting genes associated with kidney function. One suggestion here. Since the sample size of eGFRcr GWAS is twice as eGFRcys GWAS. It will be possible that some TWAS signals showed strong effects on eGFRcr but only showed marginal effects on eGFRcys due to power issue. This may help pick some of the real signals back.

One related question, does TWAS results of BUN been treated as a filtering step for kidney function associated genes? It is not directly clear what is the value of including BUN if it has not been used as a selection criteria.

Response: Given the different sample sizes, we did consider performing some sort of discovery followed by replication analysis as raised here. However, the analysis proved to be well powered even when only considering the overlap of fully multiple testing adjusted eGFRcr and eGFRcys findings. Hence, we stayed conservative and prioritized excluding potential false positives introduced by a more relaxed threshold over the potential false negatives included in our current design. For BUN, the reduction in sample size was more severe, and the loss in power was even more pronounced than indicated by sample size, possibly due to the greater physiological variation in BUN as compared to eGFR. Overall, we were encouraged when targets were identified using all three (eGFRcr, eGFRcys, and BUN), but we felt that the difference between the physiology behind GFR and BUN, compounded by the large difference in sample size, was too pronounced to interpret associations with BUN at the same level as eGFRcr and eGFRcys, and thus we focused on the direct overlap of targets between eGFRcr and eGFRcys. We included these aspects of power in the discussion (lines 386-8).

2. Power of PWAS and TWAS, pQTL data were measured in more samples. However, when comparing top TWAS and PWAS signals, why the number of PWAS signals were much less than TWAS? Is this just an issue of number of proteins been tested?

Response: Thank you for bringing this matter up. We have collected the number of associations and the respective row-wise percentages of tested transcripts/proteins in **Reviewer Table 1**. Per trait, the percentages are relatively similar with a slight advantage for kidney tubule transcripts and plasma proteins. While this might suggest that the tubule is a particularly relevant tissue or that the protein models have the best powered data source, we cannot dissect the tissue-relevance and modelling-sample-size components on this general level. Furthermore, the proteins represented in the protein models represent those measured well by the SomaScan platform and are not a random subsample of all proteins. Hence, in the manuscript we focused on conditional analyses within genomic regions instead of comparisons at the genome-wide level.

Reviewer Table 1: Associated features with kidney function and damage

Tissue \ Trait	eGFRcr GWAS N = 1,004,040	eGFRcys GWAS N = 460,826	BUN GWAS N = 243,031	ACR GWAS N = 547,361
Kidney Cortex #models=2633, eQTL N=73	88 / 3.3%	55 / 2.0%	6 / 0.3%	8 / 0.3%
Kidney Tubule #models=1875, eQTL N=121	146 / 7.8%	65 / 3.5%	16 / 0.9%	15 / 0.8%
Liver #models=5213, eQTL N=208	249 / 4.8%	113 / 2.2%	20 / 0.4%	23 / 0.4%

Whole Blood #models=9388, eQTL N=670	366 / 3.9%	183 / 1.9%	33 / 0.4%	51 / 0.5%
Plasma #models=1342, pQTL N=7,213	69 / 5.1%	41 / 3.1%	4 / 0.3%	10 / 0.7%

3. the percentage of TWAS/PWAS signals with coloc evidence is a bit lower than I expected. Is this a kidney function specific thing?

Response: The lower rate of TWAS / PWAS signals supported by colocalization in our kidney analyses could indicate that signals are clustered in genomics regions with confounding due to linkage disequilibrium. Across traits and tissues, 20% of TWAS signals were supported by colocalization (i.e., had coloc posterior probabilities (PP) >80%), which increased to 29% when subsetting to those using eGFRcr GWAS and kidney tubule models. For PWAS signals, 43% of associations were supported by colocalization. In comparison, the Hellwege, et al. ¹ eGFRcr analysis had 42% of associations colocalizing (we assume they used the more lenient COLOC default threshold of PP>50%). Looking beyond kidney, Zheng, et al. ² conducted a *cis* MR study that identified 180 associations between 65 proteins and 52 phenotypes. Of these 62% had PWCoCo coloc PP>80%. To truly answer this question, we would require a phenome-wide TWAS / PWAS analysis including well-powered kidney traits such that the MR / TWAS / PWAS and the colocalization methods and thresholds are kept constant, which is beyond the scope of this manuscript.

4. Conditional analysis, I think this section is the most value-added section in this manuscript. I roughly understand that this approach helps to identify distinct TWAS/PWAS signals in a region (nice demonstration in Figure 4). However, it is not easy to follow some of the details by reading the results and methods. Few questions:
 - a. The conditional analysis was conducted between TWAS and PWAS signals for one kidney trait? Or across multiple kidney traits. The *cis*-regulated genetic correlation is the r_g between transcript and kidney function or r_g between transcript and protein? Does the genetic correlation help for filtering out signals that need to do conditional analysis?

Response: Conditional analyses were conducted for one kidney trait at a time. Otherwise, the varying underlying GWAS source would not allow us to employ the regression with summary statistics approach.³ The incorporated *cis*-regulated genetic correlation is referring to the genetically encoded co-expression of the two conditionally analyzed models – so transcripts and proteins. Finally, instead of using the genetic correlation beforehand to filter which conditional analyses can be omitted we conducted all analyses within regions. This then

includes very clear results, such as in the illustrated example in **Figure 4**. We have revised the corresponding paragraph to improve clarity (lines 171-4).

- b. 5MB region this is a quite wide range. will it be possible to just use centralised eQTL and pQTL + 500kb each side to select an overlapped region for both eQTL and pQTL in the same genomic region?

Response: We started with a minor variation of this (gene region +500kb on each side). Genomic regions were then only merged if overlap existed. For the majority of regions this stayed close to the original width (e.g., 124 of 153 regions were of width <2MB). Apart from the MHC region only four regions exceeded 4MB. Indeed, these large regions were then broken down into independent signals by the conditional analyses as exemplified by the >5MB region on chromosome 15 that contained independent signals for SNUPN, PARP6, and NRG4 (**Supplementary Table 4**, region number 115).

5. Linkage between TWAS/PWAS and observational analysis

- a. Proteins vs CKD progression, a fair comparison will be comparing PWAS of proteins vs CKD progression with observational evidence. Here, we are just comparing PWAS signal of kidney function/damage in one time point.

Response: We fully agree that a CKD progression TWAS / PWAS is an interesting research topic. The current GWAS data for CKD progression is much smaller in sample size; thus, as a first pass we chose to focus the kidney function at a single point in time to maximize power.

- b. It is a nice idea to test association of proteins on CKD progression, but why in AASK rather than in ARIC? Since the evidence identified in AASK may reflect ancestry specific effects.

Response: Given the genetic evidence identifying the INHBC / INHBB associations is based on common variation observed in the general population, we wanted to challenge their relevance and test their observational association in a cohort with kidney disease. Furthermore, this allowed us to simultaneously address another limitation of the genetic screen – that of underrepresentation of minoritized populations (see also the second response to **Reviewer #1**). The strong effect of INHBC on progression (HR = 1.86) even after adjustment for measured GFR motivates us that we truly identified a factor relevant for a wide range of kidney function and a diverse set of participants (from EA mostly health to AA CKD).

Minor comments:

1. Abstract:

A proportion of the abstract was used to explain the alignment of PWAS/TWAS findings and literature evidence. This give me a wrong impression that the PWAS/TWAS findings were followed by animal models etc. It is not directly clear what is the value added to include all these rather than using more efforts to explain the observational analysis and conditional analysis been actually conducted in this study.

In addition, for the term “several genes”, better to put exact number here.

The term “independent association” is not easy to follow until read over the manuscript. Better to use another term, e.g. independent gene-trait and protein-trait associations in the same genomic regions or so.

Response: Many thanks. We revised the abstract accounting for above points (lines 44-51).

2. Line 98-99, cis variants can be pleiotropic too. Better to make it clear here.

Response: Thank you. We revised the paragraph (lines 80-5).

3. Line 129, list how many genes had been tested in the TWAS?

Response: We included the number of genes and proteins tested (line 89).

4. Figure 2 and 3, no colour and gene label for eGFRcys?

Response: We choose to label genes only once and only if signals were supported by eGFRcr and eGFRcys to reduce clutter and improve clarity.

5. Line 156, positive colocalization. Colocalization analysis only provides probability rather than an effect size with direction of effect. It will be better to just say strong coloc evidence rather than positive.

Response: We removed the term *positive colocalization* from the manuscript and now just use *colocalization* (PP >80%).

References:

1. Hellwege, J.N. *et al.* Mapping eGFR loci to the renal transcriptome and phenome in the VA Million Veteran Program. *Nat Commun* **10**, 3842 (2019).
2. Zheng, J. *et al.* Phenome-wide Mendelian randomization mapping the influence of the plasma proteome on complex diseases. *Nat Genet* **52**, 1122-1131 (2020).
3. Zhu, X. & Stephens, M. Bayesian Large-Scale Multiple Regression with Summary Statistics from Genome-Wide Association Studies. *Ann Appl Stat* **11**, 1561-1592 (2017).

Second round of review

Reviewer 1

This paper describes a GWAS-supported TWAS and PWAS analyses with colocalization to fine-map hits (eGFR, GFR, BUN, ACR). They do a good job responding to previous comments.

P 6, lines 90-95. This is a very long sentence that I think could be made clearer by splitting into 2 or more sentences

In figures 2 and 3, the genes are only labeled on the top panel, which makes comparisons difficult and limits the utility of those figures.

The claim of novelty for 28 regions (P 12 L 178) is difficult to understand since these data were all previously published and reported. I may not understand what test was done here to require $p < 5E-8$ for significance. Also it is stated that these novel associations were detected via reduced multiple testing which seems at odds with $p < 5E-8$. More details perhaps should be provided here.

The claim of mediation (P 17 L 251) seems reasonable but should be evaluated with a formal mediation analysis.

The discussion of the SPATA5L1/GATM locus (P21) should mention a previous study in independent samples that also observed similar patterns of associations between eGFR and eQTLs using COLOC and S-PrediXcan (PMID: 31451708, paragraph 10 of the Discussion).

The mis-match between the ancestry of the GWAS source data and the AASK cohort should be discussed. Inclusion of these data is a strength of the paper and provides important evidence.

There seems to be a missed opportunity to look at higher-order biology here using straightforward tools like FUMA. This was requested in a prior review.

Reviewer #1:

This paper describes a GWAS-supported TWAS and PWAS analyses with colocalization to fine-map hits (eGFR, GFR, BUN, ACR). They do a good job responding to previous comments.

Response: Many thanks for the positive assessment of our revision.

P 6, lines 90-95. This is a very long sentence that I think could be made clearer by splitting into 2 or more sentences

Response: We rephrased the section to improve clarity.

In figures 2 and 3, the genes are only labeled on the top panel, which makes comparisons difficult and limits the utility of those figures.

Response: Thank you for raising this point. We revised Figure 2 and 3 and now included labels on both upper and lower plots to ease comparisons.

The claim of novelty for 28 regions (P 12 L 178) is difficult to understand since these data were all previously published and reported. I may not understand what test was done here to require $p < 5E-8$ for significance. Also it is stated that these novel associations were detected via reduced multiple testing which seems at odds with $p < 5E-8$. More details perhaps should be provided here.

Response: Many thanks. We rephrased the section to improve clarity and better differentiate between the GWAS and the TWAS/PWAS tests.

The claim of mediation (P 17 L 251) seems reasonable but should be evaluated with a formal mediation analysis.

Response: As we didn't formally test the mediation, we removed the corresponding suggestion from the text.

The discussion of the SPATA5L1/GATM locus (P21) should mention a previous study in independent samples that also observed similar patterns of associations between eGFR and eQTLs using COLOC and S-PrediXcan (PMID: 31451708, paragraph 10 of the Discussion).

Response: Thank you we now include this in the Discussion.

The mis-match between the ancestry of the GWAS source data and the AASK cohort should be discussed. Inclusion of these data is a strength of the paper and provides important evidence.

Response: Thank you for raising this point. We now included it in the discussion.

There seems to be a missed opportunity to look at higher-order biology here using straightforward tools like FUMA. This was requested in a prior review.

Response: FUMA is primarily designed to work based on GWAS summary statistics, which would be upstream of our primary results that are the TWAS/PWAS summary statistics. Correspondingly the SNP2GENE function only takes GWAS summary statistics. The GENE2FUNC function could be applied with some of the prioritized gene lists identified in our analyses. However, both the *heatmap* and *tissue specificity* analyses would interfere with our already tissue specific design. This leaves the *gene sets* (enrichment analyses) as an option. Due to the already complex nature of the integrated TWAS and PWAS of kidney function and damage across different tissues, we chose to focus on individual genomic regions in the context of drug targeting rather than another system level component. While FUMA provides many useful analytical approaches, we feel like this is out of the scope of our manuscript.